# Preparation of Gum Arabic–Maltose–Pea Protein Isolate Complexes for 1−Octacosanol Microcapsule: Improved Storage Stability, Sustained Release in the Gastrointestinal Tract, and Its Effect on the Lipid Metabolism of High−Fat−Diet−Induced Obesity Mice

**DOI:** 10.3390/foods12010112

**Published:** 2022-12-26

**Authors:** Yin-Yi Ding, Yuxiang Pan, Wanyue Zhang, Yijing Sheng, Yanyun Cao, Zhenyu Gu, Qing Shen, Qingcheng Wang, Xi Chen

**Affiliations:** 1School of Food Science and Biotechnology, Zhejiang Gongshang University, Hangzhou 310018, China; 2Food Nutrition Science Centre, Zhejiang Gongshang University, Hangzhou 310018, China; 3College of Food Science & Technology, Nanchang University, Nanchang 330031, China; 4Collaborative Innovation Center of Seafood Deep Processing, Zhejiang Province Joint Key Laboratory of Aquatic Products Processing, Institute of Seafood, Zhejiang Gongshang University, Hangzhou 310018, China; 5Hangzhou Linping Hospital of Traditional Chinese Medicine, Hangzhou 311106, China; 6Zhejiang Provincial People’s Hospital, Affiliated People’s Hospital of Hangzhou Medical College, Hangzhou 310000, China

**Keywords:** 1-Octacosanol, microcapsule, storage stability, sustained release, HFD-induced obesity mice

## Abstract

1-Octacosanol (Octa) is a natural compound with several beneficial properties. However, its poor water solubility and metabolism in the digestive tract reduce its efficacy. The Octa-GA-Malt-PPI microcapsule was prepared as follows: gum Arabic (GA):maltose (Malt):pea protein isolate (PPI) = 2:1:2; core:shell = 1:7.5; emulsification temperature 70 °C; pH 9.0. An in vitro simulated gastrointestinal tract was used to analyze the digestion behavior. C57BL/6 mice were selected to establish an obesity model induced by a high-fat diet (HFD) to evaluate the effect of Octa monomer and the microcapsule. The diffusivity in water and storage stability of Octa improved after encapsulation. The microcapsule was ascribed to electrostatic interactions, hydrogen bonding, and hydrophobic interactions. The sustained release of Octa from the microcapsule was observed in a simulated gastrointestinal tract. Compared with Octa monomer, the microcapsule was more effective in alleviating the symptoms of weight gain, hypertension, and hyperlipidemia induced by HFD in mice. In conclusion, the construction of microcapsule structure can improve the dispersibility and stability of Octa in water, achieve sustained release of Octa in the gastrointestinal tract, and improve its efficiency in alleviating the effects of HFD on the body.

## 1. Introduction

1−Octacosanol (Octa) is a natural bioactive compound, which mainly exists in bee wax, wheat−germ oil, and rice bran wax [1]. It exhibits several biological activity characteristics, including cholesterol−reducing effects, cell protection, antifatigue, and anti−inflammatory effects, as well as improves physiological characteristics of energy metabolism [2,3,4,5]. As Octa is safe for human consumption, it is widely used in drug and functional food development [6]. However, applications of Octa are limited because of its poor solubility in water [7]. Additionally, in vivo and in vitro Octa metabolism is associated with fatty acid metabolism via β−oxidation, and shortened saturated and unsaturated fatty acids are formed in the gastrointestinal tract after the oral intake [8]. Therefore, we believe that, after digestion in the gastrointestinal tract, Octa is metabolized, thereby reducing its biological efficacy. A way should be found to improve Octa diffusion and stability in water and ensure its sustained release in the gastrointestinal tract to ensure its biological efficacy.

Encapsulation, which is an approach to protect the natural active substance and create delivery systems, has been shown to improve the activity of the functional substances in the gastrointestinal tract and achieve sustained release [9]. Pea protein−alginate microcapsules showed excellent protection against bacterial during storage and in a simulated gastrointestinal environment [10,11]. Previous studies have found that the use of plant proteins to construct a plant protein−Octa complex can significantly improve the diffusion and stability of Octa in water [12]. Previously studied shell materials of microcapsules often used polysaccharides, and some studies have introduced composite shell materials, which can further improve the encapsulation efficiency (EE) of the microcapsules by combining dextrins or proteins with polysaccharides [13,14]. Nevertheless, a few studies have focused on the sustained release of Octa from the complex in the gastrointestinal tract and its effect on mice with high−fat−diet (HFD)−induced obesity.

Due to high surface hydrophobicity and aggregated state, pea protein isolates (PPI) can be an appropriate choice as a nanocarrier for liposoluble bioactives than other proteins because it is a non−allergen protein [15]. Most of the PPI molecules are exist in their aggregated state, usually in nanoscale sizes. Polysaccharides can twine around protein surface to form a core−shell structure via electrostatic interactions [16]. Previous studies have shown that such interactions between proteins and polysaccharides can inhibit protein aggregation and further enhance the steric stability of the complexes [17,18]. After a protein−polysaccharide complex is formed, bioactives combine with the protein, and the polysaccharide plays a key role in a protective function [19]. Anionic polysaccharides interact with positively charged proteins through electrostatic interaction, and finally form core–shell nanoparticles [20]. These studies provide theoretical support for improving the water solubility of Octa by constructing a PPI−polysaccharide−Octa microcapsule.

In this study, we used PPI, gum Arabic (GA), and maltose (Malt) to fabricate a new complex for Octa encapsulation. Initially, the water solubility and the preliminary molecular mechanism of Octa encapsulation by GA−Malt−PPI complexes were investigated. Then, the digestion behavior of Octa monomer and its microcapsule in the simulated gastrointestinal tract was determined and discussed. Finally, we constructed an HFD−induced obesity model in C57BL/6 mice, and compared the effects of Octa on blood lipid levels in the obese mice before and after microcapsule construction.

## 2. Materials and Methods

### 2.1. Materials

Octa (with 90% purity) extracted from rice bran wax was obtained from Shengtao Biotechnology LLC (Huzhou, China). PPI, GA, and Malt were purchased from Shanghai Macklin Commercial Co., Ltd. (Shanghai, China). Pepsin, lipase, and cholic acid were purchased from Aladdin Biochemical Technology Co. Ltd. (Shanghai, China). All other reagents and chemicals were of analytical pure grade.

### 2.2. Octa−GA−Malt−PPI Microcapsule Preparation

#### 2.2.1. Emulsion Preparation

An Octa emulsion was prepared according to Li’s method with a slight modification [12]. Briefly, a GA−Malt−PPI solution was prepared as shell materials by dispersing PPI, GA, and Malt in water at 60 °C (mass ratio GA:Malt:PPI = 2:1:2). The Octa (core materials) was added to the GA−Malt−PPI solution while stirring continuously. The emulsions were homogenized twice using a homogenizer (IKA).

#### 2.2.2. pH Condition Optimization

Previous studies have confirmed that PPI contains several hydrophobic groups exposed to alkaline conditions [12]. Thus, pH conditions in this study were adjusted to the following values: 7.0, 8.0, 9.0, 10.0, and 11.0, maintained for 30 s, and subsequently adjusted to a neutral value (pH 7.0). GA−Malt−PPI ratios were set to 2:1:2.

#### 2.2.3. Temperature Condition Optimization

The temperature condition of the emulsion were adjusted to the following values: 50, 60, 70, 80 and 90 °C. GA−Malt−PPI ratios were set to 2:1:2. The pH was maintained at 9.0 during the emulsion process. After 30 s of emulsification reaction, adjust the pH to 7.0.

### 2.3. Determination of Particle Size and the ζ−Potential

The particle size in the samples were determined by dynamic light scattering using a Zetasizer Nano−S instrument (Malvern Instruments Ltd. Worcestershire, U.K.) at 25 °C.

### 2.4. Determination of the EE and Loading Amount (LA)

Octa content in the supernatant was extracted using trichloromethane [21], and subsequently measured using a gas chromatograph (Shimadzu Corporation, Kyoto, Japan).

EE was calculated using the following formula:EE (%) = (1 − M_1_ / M) × 100%(1)
where M_1_ represents the content of free Octa (mg), and M represents the total content of Octa (mg).

LA was calculated by the following formula:LA (mg/mg) = A_1_ / A(2)
where A_1_ represents the amount of encapsulated Octa (mg), and A represents total complex content (mg).

### 2.5. Octa Content Determination by Gas Chromatography (GC)

The content of Octa was analyzed by gas chromatography (GC) (Shimadzu Corporation), according to the Feng et al. method [7] with minor modifications. Briefly, experimental conditions were as follows: HP−5 column (30 m, 0.32 mm i.d., 0.25 μm film); injection volume 1.0 µL; injector temperature, 300 °C; detector temperature, 330 °C; column temperature, 240 °C, held for 1 min, subsequently raised at 20 °C/min to 300 °C, held for 10 min; nitrogen (N_2_) carrier gas at 45.0 mL/min; hydrogen (H_2_) carrier gas at 40.0 mL/min; fuel gas (air) carrier gas at 450.0 mL/min; detector, FID. For the quantification of Octa, a stock solution (10 mg/mL) was prepared by dissolving Octa in chloroform. To obtain the calibration curve, standard solutions at concentrations ranging from 0 to 2.5 mg/mL were prepared. The peak time of Octa and the calibration curve were shown in Figure 1A,B.

### 2.6. Digestion Experiment using the Simulated Gastrointestinal Tract

#### 2.6.1. Preparation of Storage Solution, Simulated Gastric Fluid (SGF) and Simulated Intestinal Fluid (SIF)

According to Liu’s [22,23] and Minekus’s [24] methods, incubate the microcapsule in SGF mimicking fasted conditions of stomach and subsequently in SIF in double jacketed reactors with continuous stirring. The incubation temperature was maintained at 37 °C. The composition of the SGF and SIF are described in Appendix A.

#### 2.6.2. Preparation of Octa−GA−Malt−PPI Stock Solution and Octa Stock Solution

The Octa−GA−Malt−PPI stock solution and Octa stock solution were prepared by dissolving Octa−GA−Malt−PPI microparticles or Octa, respectively, in water at a concentration of 10 mg/mL.

#### 2.6.3. Sustained Release of Octa from the Octa−GA−Malt−PPI Microcapsule in the Simulated Gastrointestinal Tract

A simulated gastric juice mixture was prepared by mixing the Octa−GA−Malt−PPI stock solution or Octa stock solution and SGF at a ratio of 1:3 (*v*/*v*). Subsequently, CaCl_2_ was added to it and pH was adjusted to 2.0 using 2 M HCl at 37 °C under continuous shaking for 10 min. Further, pepsin (2000 U/mL) was added and samples (8 mL) were analyzed at different time intervals (0, 10, 30, 60, and 120 min). Before gas chromatography−mass spectrometry (GC−MS) analysis, the samples were heated at 70 °C for 3 min to inactivate pepsin.

After 2 h of incubation in SGF, the pH of the solution was adjusted to 7.4 using 2 M NaOH and mixed 1:3 (*v*/*v*) with SIF. The simulated intestinal juice mixture was preheated at 37 °C for 10 min. Subsequently, cholic acid (0.02 mol/L) and lipase (260 U/mL) were added. During the entire reaction period, the pH was maintained at 7.4, and the pH of the reaction system was adjusted with an automatic pH constant meter (TIM 854, Radiometer, Copenhagen, Denmark). Samples (8 mL) were collected at different time intervals (0, 10, 30, 60, 120, and 240 min) and were heated at 70 °C for 3 min to inactivate digestive enzymes.

#### 2.6.4. The Release Rate (RR) of the Octa from Octa−GA−Malt−PPI Microparticles in the SGF or SIF

The sustained RR of Octa from the microparticles in the SGF or SIF was calculated by the following formula:RR (%) = (X_1_ / X) × 100%(3)
where X_1_ represents the amount of free Octa in the microcapsule after simulated digestion (mg), and X represents the total amount of Octa in the microcapsule without simulated digestion (mg).

#### 2.6.5. The Analysis of Digestive Products of the Octa Monomer in the Simulated Gastrointestinal Tract

To evaluate the digestion behavior of Octa in the gastrointestinal tract, the Octa monomer was first digested in the SGF for 120 min (the final concentration of Octa in SGF was 2 mg/mL) and subsequently rapidly transferred into the SIF for 120 min. Digestive products were analyzed using a GC−MS system (7890A−5975C, Agilent, CA, U.S.A.). GC conditions were as follows: DB−WAX column (60 m, 0.25 mm i.d., 0.5 µm film); injection volume, 2.0 µL; injector temperature, 300 °C; non−split injection; constant current mode; flow rate, 2 mL/min. Column temperature, 50 °C, held for 1 min, subsequently raised at 20 °C/min to 200 °C, further raised at 3 °C/min to 240 °C, held for 20 min. MS conditions were set as follows: an ion source, electron ionization; ion source temperature, 230 °C; quadrupole temperature, 150 °C; mass range, 33−500 amu. 

### 2.7. Fluorescence Spectroscopy

To study the interaction between components of the microparticles, the samples were analyzed using a fluorescence spectrophotometer (Shimadzu Corporation, Kyoto, Japan). Emission fluorescence was monitored at 300−450 nm, whereas excitation fluorescence was monitored at 280 nm, with a slit width of 10 nm.

### 2.8. Fourier Transform Infrared (FTIR) Spectra

The mass ratio of the sample to KBr was controlled as follows: sample:KBr = 1:100, using a tableting machine to press the ground mixed powder into a uniform transparent sheet. The samples were scanned 16 times at 25 °C in the wavelength range of 400−4000 cm^−1^ using an FTIR spectrometer (Thermo Scientific, Waltham, MA, USA).

### 2.9. Animal Experiment

#### 2.9.1. Animals and Sample Collections

This experiment was performed according to Animal Care and Use Committee of Animal Nutrition Institute of Zhejiang University of Technology, Chinese Guidelines for Animal Welfare and Experimental Protocol, and the 1964 Declaration of Helsinki and its amendments. Specific−pathogen−free (SPF) male C57BL/6 mice (aged 4 weeks, weighing 18 ± 1 g) were purchased from Slac Laboratory Animal Co. Ltd. (Shanghai, China) and housed in an environmentally controlled facility (20–24 °C, relative humidity 40−60%, 12 h light–dark cycle) in Experimental Animal Center of Zhejiang University of Technology (Hangzhou, China, ethical approval code: 20210930074). All mice in this experiment had free access to pure drinkable water during the 15−week experiment period.

After a 7−day acclimation, the mice were divided into 2 groups randomly: a control group (CON) (*n* = 10), which received a normal diet (20% of fat energy ratio), and an experiment group (*n* = 40), which received HFD (60% of fat energy ratio) to establish the obesity model. After 6 weeks of HFD treatment, 40 successfully modeled mice in the experiment group were divided into 4 groups randomly (*n* = 10): HFD group, which received HFD; HFD + Octa group (HFD+O), which received HFD and the daily gavage administration of Octa (20 mg/kg BW); HFD + shell material group (HFD+S), which received HFD and the daily gavage administration of the shell material (150 mg/kg BW); and HFD + microcapsule group (HFD+M), which received HFD and the daily gavage administration of Octa−GA−Malt−PPI microcapsule (170 mg/kg BW). Mice in CON group (*n* = 10) continued to be fed with the normal diet. The mice were weighed weekly.

After 15−week period, the mice were fasted overnight for 12 h and then anesthetized by 5 min isoflurane exposure. The blood sample was collected from the orbital sinus of mice into anticoagulant tubes. The whole blood in the anticoagulant tube was centrifuged (1570× *g*, 10 min, 4 °C) and the supernatant was taken as the plasma sample. Mice were sacrificed by cervical dislocation. The livers were dissected according to the morphological features and rinsed with normal saline. Approximately 0.1 g of undamaged liver sample was fixed in 10% formaldehyde phosphate buffer. Approximately 0.1 g of undamaged liver sample was immersed in optimum cutting temperature (OCT) compound (Sakura Finetech) and snap frozen with the rest of the liver samples in liquid nitrogen for further analysis.

#### 2.9.2. Blood Pressure Determination

After 9 weeks of gavage, the blood pressure was determined using the CODA Monitor Noninvasive Blood Pressure system (CODA−M1, Kent Scientific Corporation, Torrington, CT, USA).

#### 2.9.3. Lipid Metabolism Index Measurement

The levels of total cholesterol (kit A111−1−1), triglyceride (kit A110−1−1), high−density lipoprotein cholesterol (HDL−C, kit A112−1−1), and low−density lipoprotein cholesterol (LDL−C, kit A113−1−1) in the plasma were quantitatively determined using corresponding kits (Jiancheng Bioengineering Institute, Nanjing, China).

#### 2.9.4. Histopathological Analysis of Liver

For histopathological analysis, the liver sample that fixed in 10% formaldehyde phosphate buffer was embedded in paraffin. Hematoxylin and eosin (H−E) staining was performed on 5 μm serial sections to analyze the histopathological changes in the liver.

For liver fat accumulation analysis, the liver sample that was embedded in OCT compound was sectioned using a cryomicrotome (CM 1950, Leica, IL, U.S.A.). Tissue sections (5 μm) were stained with Oil Red O for lipid deposition using corresponding kit (Sigma−Aldrich, Milwaukee, St. Louis, MO, U.S.A.). Images were captured using an optical microscope (BX41, Olympus, Tokyo, Japan) and analyzed by Image−Pro Plus 5.0 software (Media Cybernetics, Inc., Singapore City, Singapore).

#### 2.9.5. Total RNA Isolation from Liver and Quantitative RT−PCR (qRT−PCR)

Total RNA of liver was extracted with Trizol reagent (Vazyme Biotech Co. Ltd., Nanjing, China). The total RNA concentration in each sample was quantified by NanoDrop Spectrophotometer (ND2000, Thermo Scientific, Waltham, MA, U.S.A.). A qRT−PCR kit based on SYBR green was employed and the reaction was performed in 7900HT instrument (Applied Biosystems, Waltham, MA, U.S.A.). The product specificity was assessed by melting curve analysis. The 2^−ΔΔCT^ method was used to determine the gene expressions. The sequence of the primers is shown in Appendix A. The relative expression of the genes was expressed as a ratio to the housekeeping gene β−actin.

### 2.10. Statistical Analysis

Each experiment was performed in triplicates. Data were analyzed using SPSS (version 22.0, IBM corporation, Armonk, NY, U.S.A.) expressed as the mean ± standard deviation. Comparisons between different groups were performed using one−way analysis of variance with Duncan’s test for post hoc analysis, and the significance level was set at *p* < 0.05.

## 3. Results and Discussion

### 3.1. The Optimization Conditions for the Octa−Loaded Core–Shell Complexes Preparation

#### 3.1.1. Condition Optimization for Emulsifying pH

As shown in Table 1, the ζ−potential absolute value of the complexes increased with an increase in pH, and the charge on the surface of the complexes was close to zero. Therefore, the electrostatic repulsion between the particles decreased, and aggregation occurred, which increased the particle size. The EE and LA of Octa by GA−Malt−PPI complexes first increased and then decreased with the increase in pH. In the present study, when the pH was 9, the values of EE and LA reached their maximum values. In strong alkaline conditions, the PPI was resolved and the strong alkalinity decreased the number of attachment sites between the PPI and Octa [13]. We also found that the EE and LA rapidly decreased when the pH was 10, which might be attributed to the PPI resolution caused by strong alkalinity.

#### 3.1.2. Condition Optimization for Emulsifying Temperature

The characterization of the complexes prepared at different emulsifying temperatures is presented in Table 2. The absolute values of the ζ−potential, EE, and LA first increased and then decreased with the increase in temperature. When the emulsifying temperature reached 70 °C, the complexes had the highest absolute values of the ζ−potential, EE, and LA. Heat treatment led to the thermal denaturation of PPI, which resulted in the unwinding of the tightly packed structure of proteins and exposed the hydrophobic groups [25]. At lower temperatures, the protein denaturation was not distinct, and the formation of the microcapsule structure was small, which could not effectively encapsulate the core material [26]. In the present study, we found that 70 °C was the optimal temperature for the preparation of the microcapsules. As the emulsifying temperature increased further, the solvent in the microcapsule preparation system evaporated rapidly, and the denaturation of the soybean protein was incomplete, which resulted in a decrease in EE.

### 3.2. Molecular Mechanism for Underlying Octa Encapsulation by GA−Malt−PPI Complexes

#### 3.2.1. The Fluorescence Quenching of the Complexes

Hydrophobic amino acid residues, such as tryptophan (Try) and tyrosine (Tyr), are the main fluorescent components of proteins. Fluorescence spectroscopy is usually used to determine the conformational change in proteins that react with hydrophobic bioactive substances, which can lead to changes in the fluorescence properties of the system [27,28]. Try fluorescence is especially sensitive to the polarity of the surrounding environment and can thus be used to determine the binding affinities of Try−containing proteins [29]. As presented in Figure 1C, a redshift of the Try absorption peak in the PPI molecule was observed after the addition of GA, which indicated a change in the secondary structure of the PPI molecule due to the binding between the PPI and GA, and an increase in the polarity of the amino acid residue in the PPI molecules [30]. The fluorescence intensity (FI) of the Octa−GA−Malt−PPI complex was lower than that of PPI and GA−Malt−PPI complex, which indicated that the interactions between Octa and PPI regularly quenched the endogenous fluorescence of the protein. The increase in pH led to an increase in the FI of the GA−Malt−PPI complex, which indicated that more Try residues were exposed in an alkaline environment that was more conducive to binding with Octa (Figure 1D). The increase in pH also led to a blue shift in the Try absorption peak in the PPI molecule, which indicated that the microenvironment of the Try residue became more hydrophobic [31]. As presented in Figure 1E, the FI of the PPI emission peak decreased regularly with an increasing Octa concentration. Therefore, we speculated that hydrophobic interactions are involved in the micro−complex system.

#### 3.2.2. The Interaction between Octa and GA−Malt−PPI Complexes by FTIR

FTIR analysis was used to elucidate the possible mechanism underlying the formation of micro−complexes among Octa, GA, Malt, and PPI (Figure 2). The band at 3100−3500 cm^−1^ was caused by the vibration of the O−H bond of the hydroxyl groups or the N−H bond [32]. The characteristic peaks of PPI were obtained by the stretching vibration of the O−H bond at approximately 3309.29 cm^−1^. The peak at approximately 2921.49 cm^−1^ was obtained by the saturated C−H stretching vibration [33]. The C=O stretching (amide I) and N−H bending (amide II) bands for PPI were observed at approximately 1652.76 cm^−1^ and 1537.29 cm^−1^, respectively [34]. The peaks at approximately 1078.88 cm^−1^ (amide III) were due to the N−H bending [35]. For GA, the FTIR spectrum had bands at approximately 1420 cm^−1^ arising from the −COO− symmetric stretching vibration [36]. Furthermore, the bands at approximately 3414.43 cm^−1^ and 2937.12 cm^−1^ corresponded to the O−H and C−H stretching vibrations, respectively. The characteristic peaks of Octa mainly included the O−H characteristic peak at approximately 3377.42 cm^−1^, sharp −CH_2_− asymmetric stretching vibration absorption peaks at approximately 2916.63 cm^−1^, and sharp −CH_3_ symmetric stretching vibration absorption peaks at approximately 2848.37 cm^−1^. Moreover, the −CH_3_ bending vibration and −CH_2_− bending vibration in Octa produced characteristic absorptions at 1462.91 cm^−1^ and 1378.58 cm^−1^, respectively [32].

PPI contains many reactive side groups, such as −NH_2_, −OH, and −SH, which can readily participate in the crosslinking reactions [37,38]. After the formation of the GA−PPI complex, the characteristic peak generated by the −COO− symmetric stretching vibration in GA was not observed, which could be due to the electrostatic interaction between −NH_2_ in PPI and −COO− in GA [39]. After the addition of GA, the peaks of the amide I and amide II bands of PPI were considerably reduced, which indicated that polysaccharides weakened the interaction between the PPI molecules. Additionally, a blueshift in the C−H stretching vibration was observed in the GA−PPI complex at approximately 2931.65 cm^−1^, which also indicated that the PPI molecular interaction was weakened. The shift in the absorption band of C=O from 1652.76 cm^−1^ to a lower wavenumber (1633.28 cm^−1^) indicated that the electron cloud density decreased, which further implied that a strong hydrogen bond was formed between the amino acids in PPI and GA [40]. Compared with PPI, the vibrational stretching strength of the GA−PPI complex increased and shifted to a high wavenumber of 3388.09 cm^−1^, which was because of the Maillard reaction between the carboxyl group at the reductive end of GA and the amino group of PPI molecule that resulted in covalent crosslinking and the formation of a new N−H bond. After the formation of the GA−Malt−PPI complex, the vibration stretching of N−H or O−H moved to a high wavenumber of 3394.80 cm^−1^, which indicated that the addition of Malt was conducive to the further covalent crosslinking between the GA and PPI [41]. Furthermore, the absorption peaks were similar to that of GA−PPI and GA−Malt−PPI in the amide I and amide II bands, which indicated that the addition of Malt did not significantly affect the binding of GA and PPI.

In contrast to PPI and GA, the −OH peaks of the GA−PPI, GA−Malt−PPI, and Octa−GA−Malt−PPI microcapsules were significantly shifted, which indicated the presence of hydrogen−bonding interactions among Octa, GA, Malt, and PPI. The characteristic peaks of Octa and GA−Malt−PPI were observed in the Octa−GA−Malt−PPI complex, which further confirmed that Octa did not affect the structure of the GA−Malt−PPI complex, and we could speculate that Octa might mainly bind to the hydrophobic surface area of PPI. A coupling effect of the symmetric stretching of −CH_2_− and the symmetric stretching vibration of −CH_3_ at approximately 2917.30 cm^−1^ and 2849.03 cm^−1^ was observed, which indicated that the stretching vibration interfered with the C−H in Octa and PPI [42]. Additionally, the characteristic peaks generated by the asymmetric stretching of −CH_2_− had a greater blue shift in the micro−complex of Octa−GA−Malt−PPI than that of the GA−Malt−PPI complex, which implied the presence of electrostatic interactions in the system. Therefore, the formation of the Octa−GA−Malt−PPI microcapsule was due to hydrogen bonding, hydrophobic interactions, and electrostatic interactions among Octa, GA, Malt, and PPI.

The changes in the secondary structure of the PPI molecule were determined by the FTIR spectrum combined with deconvolution and second derivative analysis by OMNIC software (Nicolet) and OriginPro software (Originlab). The amide I (1700−1600 cm^−1^) was mainly studied in the present study because the changes in the amide I on the protein secondary structure were most obvious. The secondary structure includes the intermolecular β−sheet (1610−1624 cm^−1^), intramolecular β−sheet (1624−1640 cm^−1^), random coil structure (1640−1650 cm^−1^), α−helix (1650−1663 cm^−1^), and β−turn (1666−1692 cm^−1^) [12]. The α−helix showed a negative correlation with surface hydrophobicity, and a lower random coil structure usually indicates a lower surface hydrophobicity [27]. As shown in Table 3, the results obtained in this study indicated the decreased random coil structure and increased α−helix of PPI with the addition of GA and Malt, which could confirm that the surface hydrophobicity was decreased. A decreased random coil and enhanced α−helix were observed in the Octa−GA−Malt−PPI microcapsule, which also confirmed that the surface hydrophobicity was decreased after the combination with the Octa.

### 3.3. Encapsulation Improved the Stability of Octa in Different Storage Periods

For the commercial applications of the Octa−GA−Malt−PPI microcapsule, the shelf life of food is very important. Therefore, they should be stable throughout their shelf life. By detecting the change in the ζ−potential, particle size, and polymer dispersity index value (PDI) during the three months of storage, we determined whether the complex had a good storage stability. As presented in Figure 3A, the ζ−potential, particle size, and PDI value fluctuated slightly over time, which indicated that the distribution of the particle size and stability were relatively uniform during three months of storage. Moreover, Octa is sparingly soluble in water during the 3−month storage time. After the construction of microcapsules, no turbidity or obvious floc was visible in the samples (Figure 3B), which suggested that the microcapsules had a high stability and were suitable for commercial use.

### 3.4. The Digestive Behavior of Octa and the Sustained Release of Octa from the Microcapsule in a Simulated Gastrointestinal Tract

The physiological conditions of the human gastrointestinal tract are extremely complex, containing various digestive enzymes and inorganic salts. The content and activity of the digestive enzymes and pH change with the intake of food [43,44]. To determine the sustained release of Octa from the Octa−GA−Malt−PPI microcapsule in the gastrointestinal tract, the microcapsule was first placed in SGF for 120 min and was then rapidly transferred into SIF for 240 min. The cumulative RR of Octa from the microcapsule was obtained at different time points. As presented in Figure 4A, the RR of Octa in SGF was very low. The RR of Octa only reached 49.21% after 120 min of simulated gastric digestion. After digestion in SIF began, the cumulative RR increased remarkably within 30 min and reached a cumulative RR of 92.48%. The cumulative RR decreased after 30 min of intestinal digestion, which might be because the released Octa was partially hydrolyzed in the simulated gastrointestinal tract and could not be detected. We then proceeded to determine the digestive behavior of the Octa monomers in the gastrointestinal tract. As shown in Figure 4B, after 120 min of digestion, approximately 13.51% and 48.10% of the Octa monomers were digested in SGF and SIF, respectively. We then analyzed the digestive products of Octa after its digestion by the simulated gastrointestinal tract using GC−MS. As illustrated in Figure 4C, a total of 11 fatty acids were detected after simulated gastrointestinal digestion. The contents of palmitic acid and octadecanoic acid were significantly higher than other fatty acids. These results indicate that the main digestive products of Octa in the simulated gastrointestinal tract are palmitic acid and octadecanoic acid. Interestingly, three monounsaturated fatty acids (trans/cis−9−oleic acid, cis−11−eicosenoic acid, and erucic acid) were also detected after the digestion of Octa. Previous studies have reported that two fatty acid desaturases (FADS1 and FADS2) in the intestinal tract of mammals are present, which mainly target branched fatty acids and odd fatty acids [45]. Recent studies have reported that such fatty acid desaturases can participate in the gastrointestinal digestion of Octa, mainly acting on the fatty acids produced by Octa digestion to produce monounsaturated fatty acids [12]. The above results further elucidated the mechanism underlying the RR of Octa, which first increased and then decreased during the intestinal digestion.

### 3.5. Octa and Octa−GA−Malt−PPI Microcapsule Improved the Lipid Metabolism in HFD−Induced Obesity Mice

As shown in Figure 5A, from the second week of experimentation, the body weight of the mice fed with HFD was significantly higher than that of the mice fed with a normal diet. From the sixth week, the body weight of mice fed with HFD exceeded 15% of that of mice fed with a normal diet. After the HFD−induced obesity model was successfully established, we intervened with Octa and its microcapsule. The body weight of the mice in the HFD+M group and the HFD+O group were significantly lower than that of the mice in the HFD group after 7 and 8 weeks of gavage, respectively. As shown in Figure 5B, the weight−loss trend of the mice in the HFD+M group was more distinct than those in the HFD+O group. As shown in Figure 5C, the body fat of mice in the HFD+M group and the HFD+O group were significantly lower than that of the mice in the HFD group. The effect of the microcapsule was better than that of the Octa monomer.

In addition, both the microcapsule and Octa monomer significantly reduced the abnormal increase in systolic blood pressure and diastolic blood pressure induced by HFD in mice. The microcapsule showed a better blood−pressure−lowering effect (Figure 5D,E). As shown in Figure 5F−I, the mice in the HFD group showed significantly increased levels of total cholesterol, triglyceride, and LDL−C concentration, and significantly decreased levels of HDL−C levels. Both the monomer and microcapsule significantly reduced dyslipidemia caused by HFD in mice, and the effect of the microcapsule on lowering blood lipids was better than that of the Octa monomer.

Additionally, H−E staining (Figure 6A) and Oil Red O staining (Figure 6B) of liver sections of mice confirmed that the HFD mice had observable hepatic steatosis. The Octa monomer and its microcapsule effectively reduced fatty infiltration in the liver induced by HFD, and the microcapsule had a better effect.

Furthermore, the mRNA expression of genes related to lipid metabolism, including glycerol kinase (GK), Srebp1c, Fas, and Acc1, was significant altered in HFD−treated mice (Figure 6C−F). The protein encoded by the GK gene is a key enzyme that regulates glycerin uptake and metabolism. It controls the phosphorylation of glycerol by ATP to produce ADP and glycerol 3−phosphate [46]. Serbp−1c plays an important role in lipogenesis and catalyzes the transcription of fatty−acid−synthesis−related genes, such as Acc1 and Fas [47]. Fas is a critical enzyme that regulates the fatty acid synthesis rate [48]. Acc1 is a rate−limiting enzyme in the long−chain fatty acids biogenesis, and controls the fatty acid synthesis [49]. In the present study, we found that the changes in the mRNA expression of these lipid-metabolism−related genes of HFD−treated mice were effectively mitigated under administration by Octa or its microcapsule. The above results confirmed that Octa had an increased effect on reducing excessive weight gain, hypertension, and hyperlipidemia caused by HFD after it was encapsulated with PPI, whereas the shell material had no significant effect on the above indexes.

## 4. Conclusions

Octa is a natural bioactive substance. Due to its poor water solubility, its application in food product research and development is limited. In addition, we found that Octa was digested into short−chain alkanes or alkenes after gastrointestinal digestion, which also limited its biological function efficiency. In order to solve the above problems, GA−Malt−PPI complexes were constructed to encapsulate Octa. The formation mechanism of the Octa−GA−Malt−PPI microcapsule was due to the hydrogen bonding, hydrophobic interactions, and electrostatic interactions among Octa, GA, Malt, and PPI. After encapsulation, the dispersion of Octa in water improved greatly. The microcapsules had a better storage stability and achieved an ideal sustained release in the simulated gastrointestinal tract. The moderating effect of Octa on the effects of HFD was significantly increased after it was encapsulated by GA−Malt−PPI complexes. The present study showed that these microcapsules can be used in different functional food and beverages to increase the dispersion and storage stability in the food matrix and achieve a sustained release and efficient fat reduction in Octa.

## Figures and Tables

**Figure 1 foods-12-00112-f001:**
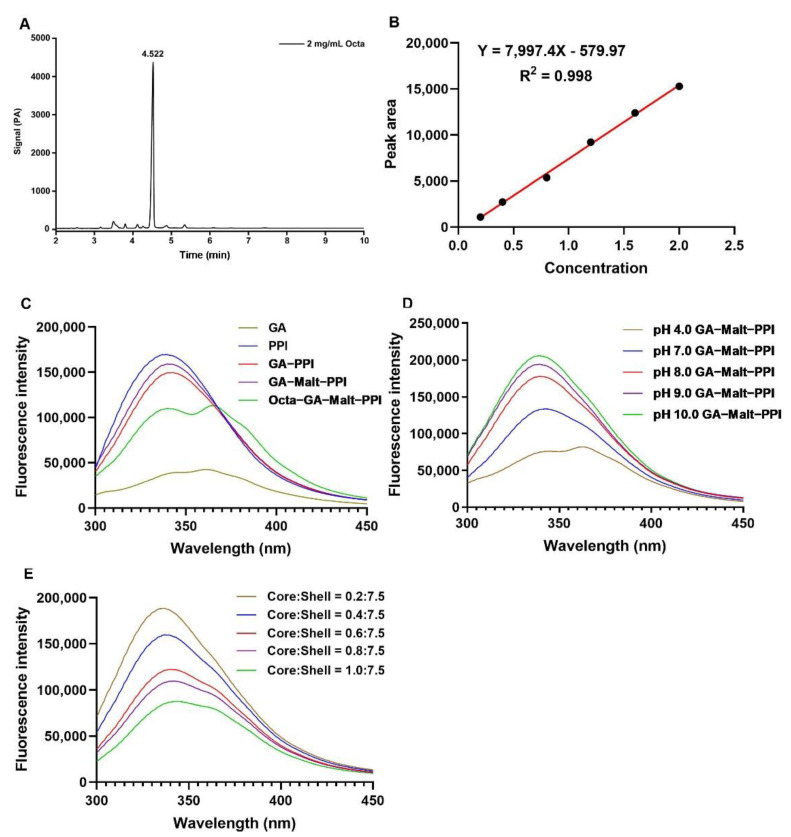
(**A**) Peak time for Octa in GC analysis. (**B**) Calibration curve of Octa in GC analysis. (**CE**) Fluorescence emission spectra of samples. (**C**) GA, PPI, GA−PPI, GA−Malt−PPI, and Octa−GA−Malt−PPI. (**D**) GA−Malt−PPI at different pH. (**E**) Octa−GA−Malt−PPI at different core:shell ratios.

**Figure 2 foods-12-00112-f002:**
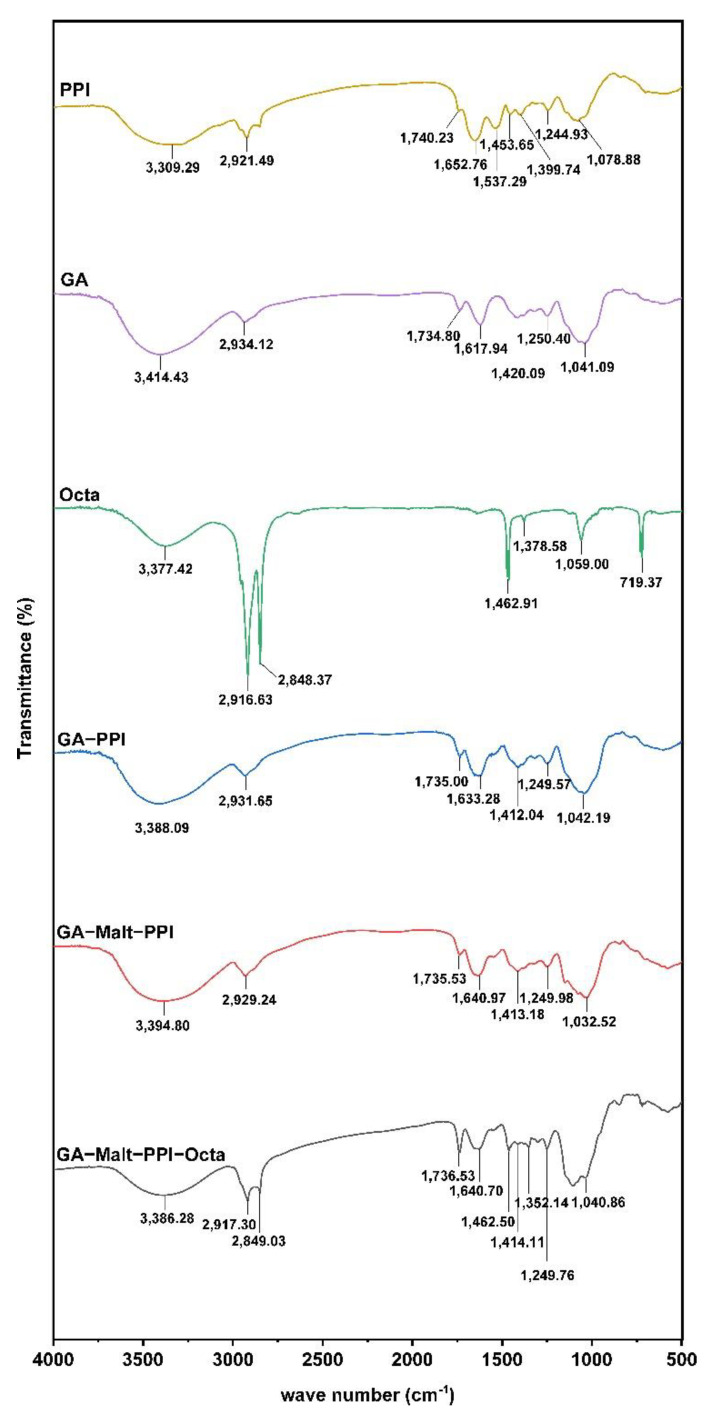
FTIR spectra of PPI, GA, Octa, GA−PPI, GA−Malt−PPI, and Octa−GA−Malt−PPI.

**Figure 3 foods-12-00112-f003:**
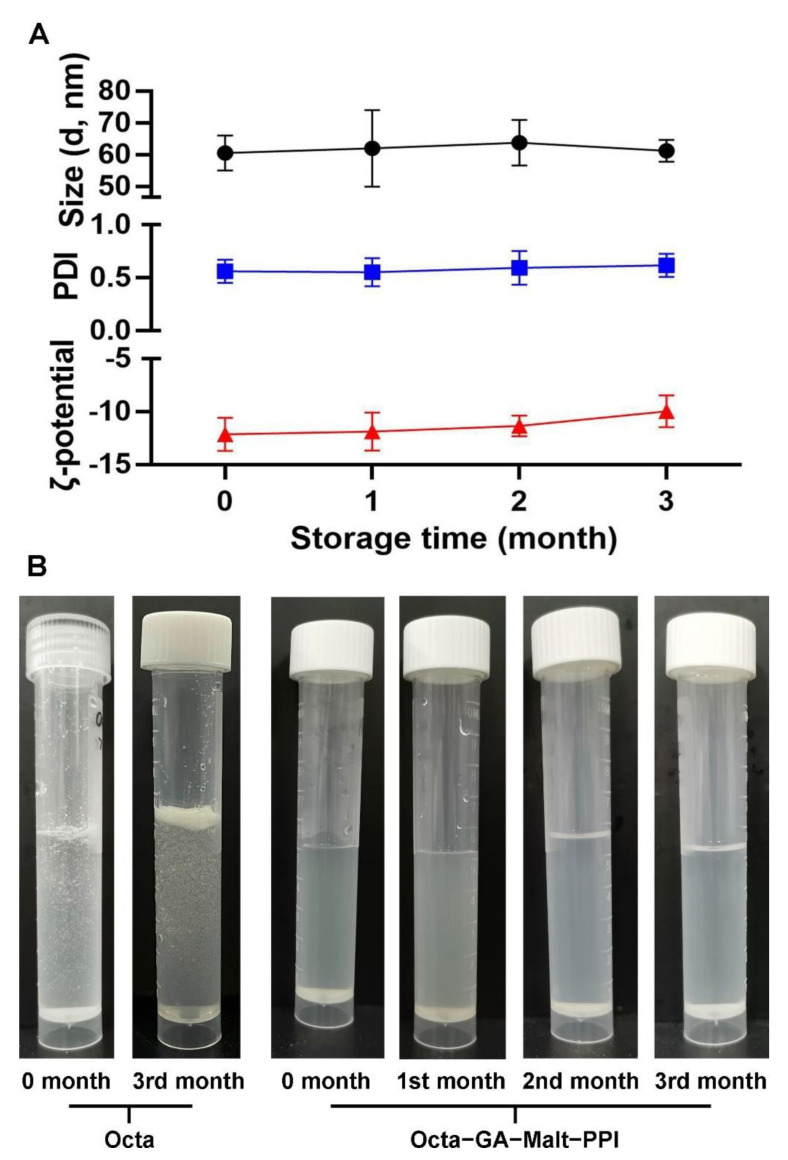
Effect of different storage times on the stability of Octa−GA−Malt−PPI microcapsule. (**A**) ζ−potential, PDI, and particle size. (**B**) The images of the Octa monomer and Octa−GA−Malt−PPI microcapsule during the 3−month storage period.

**Figure 4 foods-12-00112-f004:**
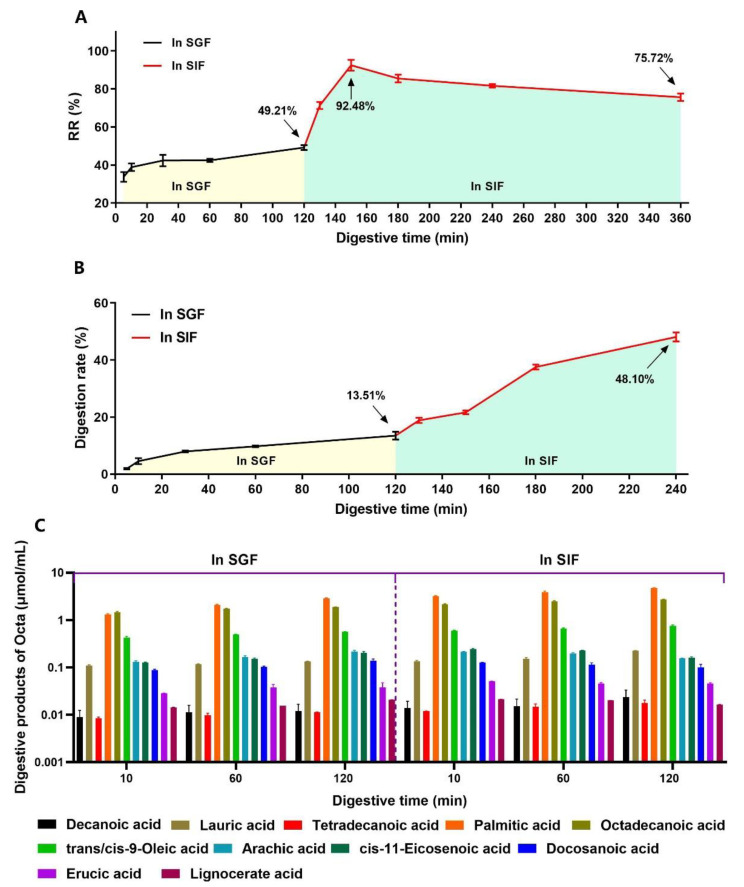
The digestion and the sustained release of Octa in simulated gastrointestinal tract. (**A**) The release rate of Octa from Octa−GA−Malt−PPI microcapsule in in vitro simulated gastrointestinal tract. (**B**) The digestive rate of Octa in in vitro simulated gastrointestinal tract. (**C**) Digestive products of Octa monomer in simulated gastrointestinal tract. The same color indicates the same digestive product.

**Figure 5 foods-12-00112-f005:**
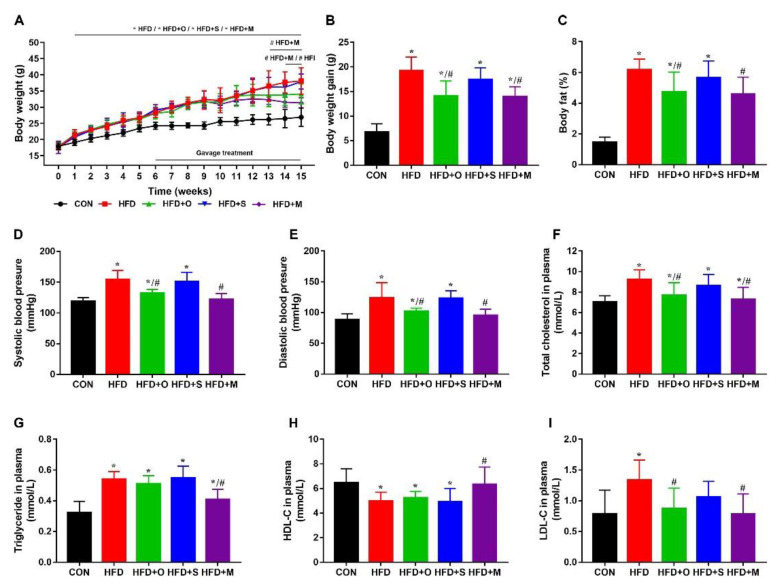
The effect of Octa monomer and the microcapsule on the HFD−induced obesity model in C57BL/6 mice. (CON: control group; HFD: obesity model group; HFD+O: HFD + Octa group; HFD+S: HFD + shell material group; HFD+M: HFD + microcapsule group) (**A**) The body weight growth of mice. The curves show the body weight growth during the experimental period. (**B**) The body weight gain after the 15−week experimental period. (**C**) The body fat percentage of mice. (**D**,**E**) The blood pressure (**D**: systolic blood pressure; E: diastolic blood pressure). (**F**) Total cholesterol in plasma. (**G**) Triglyceride in plasma. (**H**) HDL−C in plasma. (**I**) LDL−C in plasma. * *p* < 0.05, when compared with the CON group. # *p* < 0.05, when compared with the HFD group.

**Figure 6 foods-12-00112-f006:**
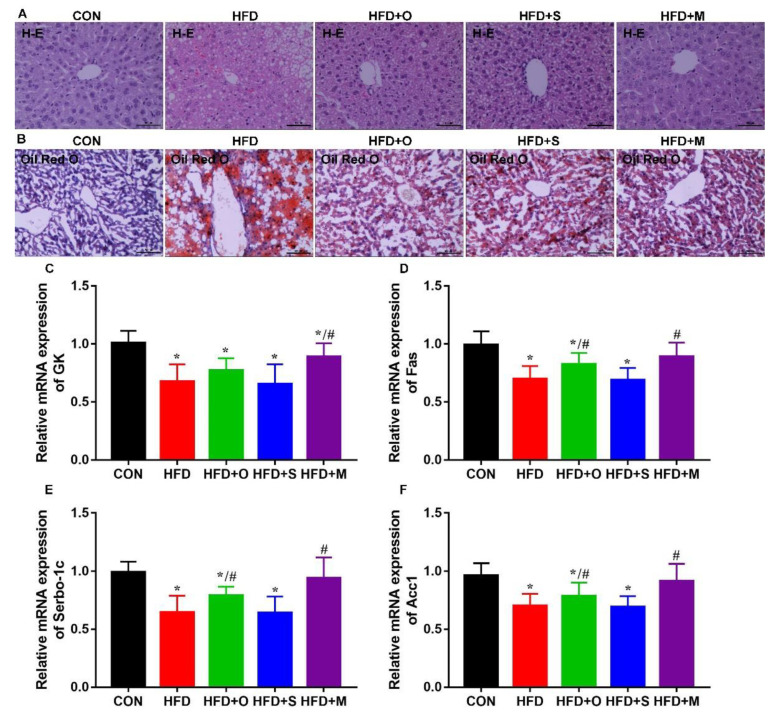
Effects of Octa monomer and microcapsule on hepatic lipid metabolism. (**A**) H−E staining (200 ×, scale bar = 100 μm). (**B**) Oil Red O staining (100 ×, scale bar = 200 μm). (**C**–**F**) The mRNA expression of lipid−metabolism−related genes in liver: (**C**) GK. (**D**) Fas. (**E**) Srebp1c. (**F**) Acc1. * *p* < 0.05, when compared with the CON group. # *p* < 0.05, when compared with the HFD group.

**Table 1 foods-12-00112-t001:** Characteristics of the microcapsule under different emulsifying pH.

pH	ζ Potential (mV)	EE (%)	LA (mg/mg)	Z−Average Diameter (μm)	PDI
7.0	−25.17 ± 1.08 ^a^	40.43 ± 3.68 ^a^	0.33 ± 0.06 ^a^	80.30 ± 6.24 ^a^	0.70 ± 0.07 ^a^
8.0	−25.43 ± 3.59 ^a^	47.67 ± 3.46 ^a^	0.41 ± 0.05 ^ab^	81.77 ± 0.2 ^a^	0.68 ± 0.01 ^a^
9.0	−31.37 ± 1.97 ^b^	58.07 ± 2.36 ^b^	0.51 ± 0.04 ^b^	138.27 ± 1.81 ^c^	0.56 ± 0.05 ^b^
10.0	−28.53 ± 1.21 ^ab^	55.90 ± 2.01 ^b^	0.45 ± 0.04 ^b^	131.83 ± 6.27 ^bc^	0.63 ± 0.03 ^ab^
11.0	−28.87 ± 1.90 ^ab^	47.70 ± 2.19 ^a^	0.40 ± 0.04 ^ab^	763.47 ± 20.06 ^d^	0.64 ± 0.03 ^ab^

Different letters: ^a^, ^b^, and ^c^ within the same column indicate statistically significant differences at *p* < 0.05.

**Table 2 foods-12-00112-t002:** Characteristics of the microcapsule under different emulsifying temperature.

Temperature	ζ Potential (mV)	EE (%)	LA (mg/mg)	Z−Average Diameter (μm)	PDI
50 °C	−20.17 ± 1.00 ^a^	56.02 ± 2.00 ^a^	0.36 ± 0.02 ^ab^	318.90 ± 3.57 ^a^	0.29 ± 0.01 ^a^
60 °C	−21.70 ± 1.00 ^a^	56.25 ± 1.68 ^a^	0.36 ± 0.04 ^ab^	291.43 ± 3.28 ^b^	0.30 ± 0.01 ^a^
70 °C	−22.37 ± 0.67 ^a^	60.99 ± 5.98 ^a^	0.48 ± 0.07 ^b^	259.47 ± 4.76 ^c^	0.24 ± 0.02 ^b^
80 °C	−16.60 ± 0.87 ^b^	51.02 ± 5.92 ^a^	0.45 ± 0.05 ^b^	273.70 ± 5.20 ^d^	0.35 ± 0.01 ^c^
90 °C	−10.08 ± 0.58 ^c^	34.10 ± 4.63 ^b^	0.28 ± 0.04 ^a^	86.10 ± 1.45 ^e^	0.50 ± 0.01 ^d^

Different letters: ^a^, ^b^, ^c^, ^d^, and ^e^ within the same column indicate statistically significant differences at *p* < 0.05.

**Table 3 foods-12-00112-t003:** FTIR spectra of PPI, GA−PPI, GA−Malt−PPI, and Octa−GA−Malt−PPI.

	PPI	GA−PPI	GA−Malt−PPI	Octa−GA−Malt−PPI
β−sheet (%)	34.07 ± 1.39 ^c^	37.57 ± 1.00 ^b^	39.66 ± 0.62 ^a^	39.37 ± 1.27 ^b^
Random coil (%)	14.29 ± 2.28 ^ab^	11.18 ± 1.16 ^b^	7.36 ± 0.67 ^c^	7.32 ± 0.75 ^a^
α−helix (%)	18.53 ± 2.80 ^b^	23.64 ± 1.48 ^a^	24.27 ± 0.15 ^a^	26.38 ± 1.63 ^b^
β−turn (%)	33.11 ± 1.19 ^a^	27.60 ± 1.33 ^b^	27.89 ± 0.37 ^b^	27.68 ± 0.64 ^b^

Different letters: ^a^, ^b^, and ^c^ within the same row indicate statistically significant differences at *p* < 0.05.

## Data Availability

Data is contained within the article or Appendix A.

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
