# Peer review of "Preparation of Gum Arabic–Maltose–Pea Protein Isolate Complexes for 1−Octacosanol Microcapsule: Improved Storage Stability, Sustained Release in the Gastrointestinal Tract, and Its Effect on the Lipid Metabolism of High−Fat−Diet−Induced Obesity Mice"

_foods, 2022, doi:10.3390/foods12010112_

Round 1

Reviewer 1 Report

The study tested the effects of microcapsulation of octacosanol using gum arbic, maltose, and pea protein isolate on their storage stability, gastrointestinal behaviors, metabolic phenotypes in C57BL/6 mice. Although octacosanols have a variety of biological activities, their low stability and poor solubility. The current study is meaningful in that they suggested a potential method to overcome the limitations of octacosanol applications. Especially, the authors analyzed characteristics of the microcapsule depending on experimental conditions for emulsifying temperature and tried to reveal potential molecular mechanisms for octa encapsulation. However, minor revisions are necessary in this journal. Specific comments are as follows;

1)      More specific information is necessary for the method section (which kit do they use for lipid metabolism index). Also, it will be helpful to understand important results if the manuscript includes more detailed information about digestion experiments using the simulated gastrointestinal tract. For statistics, what kind of post hoc tests were used to compare significance among groups?

2)      In figure 3, it is better to mention the period of storage for Octa monomer similarly with encapsulated Octa. Also, better to include the visible condition of Octa monomer as time passed similarly with the encapsulated Octa.

3)      In figure 5, Not easy to interpret the data due to abbreviations. it is better to mention in the figure legends what the abbreviations (HFD+O, HFD+S, etc) stand for?  

4)      In figure 5a, the green line represents the HFD+S group, but in figures 5A-I, the green bar represents the HFD+O group. It would be better to be consistent for the colors of the groups

5)      It is better to add more discussion about encapsulation, and other application of the encapsulation method used in the manuscript (e.g. the application of the current method for other compounds, etc)

Author Response

Point 1: More specific information is necessary for the method section (which kit do they use for lipid metabolism index). Also, it will be helpful to understand important results if the manuscript includes more detailed information about digestion experiments using the simulated gastrointestinal tract. For statistics, what kind of post hoc tests were used to compare significance among groups?

Response 1: We all agree with the reviewer’s suggestion. We apologize for the difficulty of understanding caused by our negligence. In the revised version, we describe the specific information of the kit used to determine the lipid metabolism indicators. Please check section 2.9.3. (line 221-224). Also, more detailed information about digestion experiments using the simulated gastrointestinal tract is included in the revised version. Please check section 2.6.1 and 2.6.3. (line 130-134 and line 148-154). For statistics, comparisons between different groups were done using one-way analysis of variance with Duncan’s test for post-hoc analysis, and the significance level was set at p < 0.05. Please check section 2.10. (246-248).

Point 2: In figure 3, it is better to mention the period of storage for Octa monomer similarly with encapsulated Octa. Also, better to include the visible condition of Octa monomer as time passed similarly with the encapsulated Octa.

Response 2: We all agree with the reviewer’s suggestion. However, because Octa is almost insoluble in water, it is impossible to detect the changes of ζ potential, PDI, particle size and other indicators of Octa monomer during the 3-month storage period. Also, unfortunately, we only took pictures of Octa monomer when it was first put into water (0 month) and after 3 months of storage. As shown in Figure 3B in the revised version, Octa is sparingly solubility in water during the 3-month storage time. Please check section 3.3 (line 387) and Figure 3B.

Point 3: In figure 5, not easy to interpret the data due to abbreviations. It is better to mention in the figure legends what the abbreviations (HFD+O, HFD+S, etc) stand for? 

Response 3: We apologize for the difficulty of understanding caused by our negligence. We have explained the abbreviations for each experimental group in section 2.9.1. In addition, In addition, we annotate the name abbreviations of each experimental group in the figure legend of Figure 5 in the revised version.

Point 4: In figure 5a, the green line represents the HFD+S group, but in figures 5A-I, the green bar represents the HFD+O group. It would be better to be consistent for the colors of the groups

Response 4: We apologize for the difficulty of understanding caused by our negligence. In the revised version, the colors of the groups has been consistent. Please check Figure 5.

Point 5: It is better to add more discussion about encapsulation, and other application of the encapsulation method used in the manuscript (e.g. the application of the current method for other compounds, etc).

Response 5: We all agree with the reviewer’s suggestion. In the revised version, we have discussed the application of the encapsulation method. Please check line 50-56.

Reviewer 2 Report

Very interesting paper with many interesting methods connecting in-vitro and in-vivo experiments. The Authors found, that compared with Octa monomer, designed microcapsules were more effective in alleviating the symptoms of weight gain, hypertension, and hyperlipidemia in mice. Microcapsules structure improved the dispersibility and stability of Octa in water and its release in the gastrointestinal tract. In effect its improved efficiency in alleviating the effects of HFD on the body has been proven. English language is proper. The structure of the paper is correct.

The following papers could be mentioned in the Introduction:

Zahra Khoshdouni Farahani, Mohammad Mousavi, Mahdi Seyedain Ardebili, Hossein Bakhoda, The effects of Ziziphus jujuba extract-based sodium alginate and proteins (whey and pea) beads on characteristics of functional beverage, Journal of Food Measurement and Characterization, 10.1007/s11694-022-01353-x, (2022).

Natallia Varankovich, Maria F. Martinez, Michael T. Nickerson, Darren R. Korber, Survival of probiotics in pea protein-alginate microcapsules with or without chitosan coating during storage and in a simulated gastrointestinal environment, Food Science and Biotechnology, 10.1007/s10068-017-0025-226, 1, (189-194), (2017).

J. Wang, M.T. Nickerson, N.H. Low, A.G. Van Kessel, Efficacy of pea protein isolate–alginate encapsulation on viability of a probiotic bacterium in the porcine digestive tract, Canadian Journal of Animal Science, 10.1139/cjas-2016-0090, (214-222), (2016).

Figure 6 C-F….The charts are a little too small and it is difficult to read statistics.

Author Response

Point 1: The following papers could be mentioned in the Introduction:

Zahra Khoshdouni Farahani, Mohammad Mousavi, Mahdi Seyedain Ardebili, Hossein Bakhoda, The effects of Ziziphus jujuba extract-based sodium alginate and proteins (whey and pea) beads on characteristics of functional beverage, Journal of Food Measurement and Characterization, 10.1007/s11694-022-01353-x, (2022).

Natallia Varankovich, Maria F. Martinez, Michael T. Nickerson, Darren R. Korber, Survival of probiotics in pea protein-alginate microcapsules with or without chitosan coating during storage and in a simulated gastrointestinal environment, Food Science and Biotechnology, 10.1007/s10068-017-0025-2, 26, 1, (189-194), (2017).

  1. Wang, M.T. Nickerson, N.H. Low, A.G. Van Kessel, Efficacy of pea protein isolate–alginate encapsulation on viability of a probiotic bacterium in the porcine digestive tract, Canadian Journal of Animal Science, 10.1139/cjas-2016-0090, (214-222), (2016).

Response 1: We all agree with the reviewer’s suggestion. All the above papers discussed the application and related mechanism of microcapsule system. This is something that was missing in previous versions of this article. In the revised version, the practical application of the microcapsule system is discussed in the Introduction, and the above references are cited. Please check reference 9, 11, 12. The numbering of other references has also been modified accordingly.

Point 2: Figure 6 C-F….The charts are a little too small and it is difficult to read statistics.

Response 2: We apologize for the difficulty of understanding caused by our negligence. In the revised version, we readjusted the layout of Figure 6C-F. We believe that Figure 6C-F is clear enough to read statistics in the revised version.

Reviewer 3 Report

Preparation of gum Arabic-maltose-pea protein isolate complexes for 1-Octacosanol microcapsule: Improved storage stability, sustained release in the gastrointestinal tract, and its effect on the lipid metabolism of high-fat-diet-induced obesity mice

Yin-Yi Ding, Yuxiang Pan, Wanyue Zhang, Yijing Sheng, Yanyun Cao, Zhenyu Gu, Qing Shen Tao Xu, Qingcheng Wang and Xi Chen

The work is written in an interesting way, the discussion was conducted on the basis of the results obtained from many complementary methods. It has great application potential. I found only minor errors marked in color in the text.

In my opinion, greater care should be taken when it comes to the English language, preferably with the participation of a native speaker. Please carefully consider the titles of chapters and subsections, because sometimes they are too general.

If I can suggest a literature search, sometimes it would be useful to swap to newer items, sometimes compare with similar research, more not Asian authors (science knows no boundaries), for example:

„Effect of temperature on n-tetradecane emulsion in the presence of phospholipid DPPC and enzyme lipase or phospholipase A2”, Langmuir 24(14) (2008)  7413-7420 

Edible films made from blends of gelatin and polysaccharide-based emulsifiers - A comparative study,  Food Hydrocolloids 96 (2019) 555-567 IF 5,089

"Zeta potential and droplet size of n-tetradecane/ethanol (protein) emulsions”

B: Biointerfaces

Colloids and Surfaces 25 (2002) 55-67

Release kinetics and antimicrobial properties of the potassium sorbate-loaded edible films made from pullulan, gelatin and their blends,  Food Hydrocolloids 101(2020) 105539 140

I recommend publishing this work after minor revision.

Author Response

Point 1: In my opinion, greater care should be taken when it comes to the English language, preferably with the participation of a native speaker. Please carefully consider the titles of chapters and subsections, because sometimes they are too general.

Response 1: We all agree with the reviewer’s suggestion. We have submitted our manuscript to some native speakers to improve the language of the article. In addition, we modified some of the titles of chapters and subsections that were to general. Please check 2.5, 2.9.3, 2.9.4, 2.9.5, 3.3, 3.5.

Point 2: If I can suggest a literature search, sometimes it would be useful to swap to newer items, sometimes compare with similar research, more not Asian authors (science knows no boundaries), for example:

(1) Effect of temperature on n-tetradecane emulsion in the presence of phospholipid DPPC and enzyme lipase or phospholipase A2”, Langmuir 24(14) (2008) 7413-7420

(2) Edible films made from blends of gelatin and polysaccharide-based emulsifiers - A comparative study, Food Hydrocolloids 96 (2019) 555-567 IF 5,089

(3) Zeta potential and droplet size of n-tetradecane/ethanol (protein) emulsions”B: Biointerfaces Colloids and Surfaces 25 (2002) 55-67

(4) Release kinetics and antimicrobial properties of the potassium sorbate-loaded edible films made from pullulan, gelatin and their blends, Food Hydrocolloids 101(2020) 105539 140

Response 2: We all agree with the reviewer’s suggestion. In the revised version, we cited the above article. Please check reference 10 (line 52), reference 16 (line 59), reference 27 (line 255), reference 29 (line 272).
